# Cross-Cultural Comparison of Relationships between Empathy and Implicit Theories of Emotions (in Chinese and Russians)

**DOI:** 10.3390/bs11100137

**Published:** 2021-10-11

**Authors:** Tatiana Kornilova, Qiuqi Zhou

**Affiliations:** General Psychology Department, Lomonosov Moscow State University, Mokhovaya Street, 11/9, 125009 Moscow, Russia; zhouqiuqi3@gmail.com

**Keywords:** latent class analysis, empathy, implicit theories of emotion, Questionnaire of Cognitive and Affective Empathy (QCAE)

## Abstract

The current manuscript presents the results of a cross-cultural comparison of the relationships between empathy and implicit theories of emotion in individuals from China and Russia. We hypothesized that the members of the Chinese culture would differ from the more Western Russian participants in terms of relationships between the various components of the emotional domain. Thus, we aimed to identify latent personality profiles while hypothesizing that the Chinese sample would demonstrate more prominent links between empathy and implicit theories regarding the possibility of controlling emotions. We also assumed that immediate social context could affect the results, and therefore, we compare two groups of Chinese participants—those living in China and those living in Russia, predominantly studying in Russian universities. The initial sample included Russians (N = 523), Chinese living in Russia (N = 376), and Chinese living in China (N = 423). However, following matching procedures to enable the sociodemographic comparability of samples, the final comparison was reduced to a final sample of Russians (N = 400), a sample of Chinese living in Russia (N = 363), and a sample of Chinese living in China (N = 421). We used latent class analysis and correlation analyses to test the study hypotheses. The study found that, unlike Russians, the Chinese participants demonstrated a positive correlation between incremental implicit theories of emotions and empathy. We also established significant group and gender differences. Russian women reported higher affective empathy than men, whereas Chinese women demonstrated higher affective empathy and cognitive empathy, as well as incremental implicit theories of emotion.

## 1. Introduction

Cross-cultural psychology traditionally views individualism–collectivism as one of the most consequential trait continuums that differentiate cultures [1]. Individualistic cultures emphasize social values of autonomy and independence, agency and self-regulation, and personal achievement. Collectivistic cultures (e.g., represented by Asian countries such as Japan, China, and South Korea), on the other hand, particularly value relationships among community members, social harmony, intragroup goals, and social responsibility [2]. Of particular note, however, is that cultural values are susceptible to change, which is at least partially attributed to globalization and related undercurrents of inter-cultural influence and enrichment. For example, China has recently been noted for the growing role of individualism in its culture and society [3].

Concerned with the benefits for the society and focused on relationships with others, individuals in collectivistic cultures are thought to rely on interdependent self-construal; interdependence assumes that the Self is a process that is interconnected with furthering social relationships and is guided by a realization that our behavior is significantly affected by the thoughts, feelings, and behaviors of others [4]. For example, in an interesting study, Wang and Conway [5] showed that when American participants were asked to recall 20 events, they frequently provided responses detailing memories of individual experiences and unique, singular instances and events. However, Chinese participants were likely to recall social and historical events of global value and high social importance and detailed social interactions and significant individuals in their lives.

Modern Russia’s position on the collectivism–individualism continuum is still hotly debated. In the 30 years following the demise of the Soviet Union, the previously proclaimed values of the individual have gradually replaced collectivist values. According to the traditional individualism–collectivism scale, we should assume a great degree of individualism in Russia and collectivism in China, which follows from the Hofstede Group data: i.e., China and Russia were shown to score 20 and 39 out of 100 points on individualism, respectively [6].

In a centuries-long dispute between “Westerners” and “Slavophiles” (who insist on the originality and specificity of the “Russian cultural way” as compared with the general West), the two sides agreed that Russians are characterized by a great degree of emotional responsiveness compared with the generalized image representative of “Western” cultures.

Studies in the past several decades have established that cultures differ in how individuals experience emotions [7,8], manifest and utilize emotional intelligence [9], express their attitudes towards negative life experiences [10], and even regulate prosocial behavior [11,12], which is especially pertinent in the times of the COVID-19 pandemic [13,14]. Miyamoto and Ma [8] showed that individuals from Eastern and Western cultures recruit different emotional regulation strategies when faced with positive emotions. Individuals from Eastern cultures are more likely to suppress positive effects because of dialectical thinking, reflecting a dynamic view on state development. Although empathy is frequently discussed as a component of emotional regulation, studies of its role in the context of different countries (in general and with respect to the collectivism–individualism continuum) are currently lacking. Of particular importance in this context is the investigation the relationship between empathy and how people perceive their emotion regulation skills in different cultures.

*We hypothesized that emotional regulation of behavior is influenced substantially by cultural factors, in particular with respect to empathy as well as implicit theories concerning the ability to regulate and modify one’s emotions*.

The concept of implicit theories of emotion implicates the understanding of one’s emotions and the perceived ability to control them. Eastern cultures place emphasis on interdependency and interrelatedness between people, leading to a greater need for control in emotional self-regulation, more empathy, and acceptance of beliefs regarding the malleability and manageability of emotions. Chinese society is collectivist and traditional, with a profound and long history; the person is included in it as the sum of their social roles and can realize themselves necessarily within the “magnetic field of human attachment” [15].

The current study aimed to test the hypothesis that Chinese individuals were likely to show higher levels of empathy compared to Russians. However, there is no reason to assume the same premises is true for incremental theories of emotions. We argue that cross-cultural differences in these variables should be productively examined through the lens of latent classes that account for individual differences and patterns of interrelationships between variables while constructing empirical typologies that link interindividiual variation with the presence of relatively homogenous subgroups of individuals that share similar personality profiles and patterns of relationships between them.

### 1.1. Empathy

Empathy is broadly viewed as a complex and multifactorial trait [16] that implicates cognitive, emotional, and behavioral systems, as well as moral reasoning [17], as both a trait and a state [16,18], as well as a specific ability [19]. Hoffman [20] considered empathy akin to intellectual ability, whereas others (e.g., [21]) consider empathy to be a dynamic and directed psychological process that can take a variety of shapes, from emotional resonance to emotional self-regulation. The development of the idea of unity between intelligence and affect as well as the conceptualization of higher functions in cultural-historical psychology based on L. Vygotsky’s approach enabled us to introduce new aspects in the understanding of empathy: its regulation by a multitude of processes [22] and interiorization of mono- and poly-role relations of the individual in society [23]. Thus, an empathic result can be achieved both by a more “natural” process of “identification” and by a mediated “modeling” process (a “higher” mental function, in Vygotsky’s terminology).

A series of recent observational studies directly examined the relationships between empathy and other components of emotional and personality domains: e.g., emotional intelligence [24,25], emotional regulation [26,27], and resilience [27]. Interestingly, empathy is frequently viewed as a component of emotional intelligence, and if the latter is viewed as a non-cognitive ability, so is the former [28]. Emotional regulation refers to processes that involve experiencing emotions, expressing emotions, and the capability of changing them [29]. Emotional regulation deficits are associated with low emotional intelligence [30]; empathy is also typically lower in those individuals who score highly on the Dark Triad measures [30] and higher in those individuals with prosocial personalities [31]. This is consistent with the idea that observing others’ pain furthers empathic concern and motivates prosocial action [32,33].

Empathy is related to a wide range of traits: It has been shown to be positively correlated with such Big Five traits as Conscientiousness, Agreeableness, and Openness [34]. An emergent understanding of empathy as a complex trait includes cognitive as well as emotional components [35,36,37]. Cognitive components include the ability to evaluate one’s own and others’ motivation and behavior, as well the ability to understand others and to explain and predict real-world behavior [38].

In Asian countries where the culture emphasizes each person’s involvement in mutually sympathetic social relationships, empathy might have a higher value than in other countries. At the same time, the relationships of empathy with other components of emotional-personal potential can be strikingly parallel in different cultures [39], which are perceived to have similar differences in location on the Hofstede collectivism–individualism continuum (weblink 1).

The development of programmatic cross-cultural studies is heavily tied to the development of the methodological toolbox available to researchers and practitioners alike and rooted in the culturally sensitive adaptation and validation of psychological assessments. For example, the Questionnaire of Cognitive and Affective Empathy has been successfully adapted for use in China and Russia [22,40,41] and was used in the current study reported in this manuscript.

### 1.2. Implicit Theories of Emotions

Implicit theories of emotions (ITE) are a much less studied domain of individual differences and cross-cultural differences compared to empathy or the widely popularized concept of emotional intelligence. Analogously to implicit theories of intelligence and personality [42,43,44], ITE should be viewed as an emergent system of beliefs regarding one’s or others’ traits that one develops via individual and culturally contextualized experiences [45]. Compared to social beliefs, implicit theories, as a mindset, are less accessible to consciousness.

Dweck’s original differentiation between incremental and constant theories was adapted to the emotional domain by Tamir and colleagues [45], who demonstrated that people indeed hold a system of beliefs about emotions, with a key component related to one’s perceived ability to control them. Implicit theories of emotions are viewed analogously to theories of ability: incremental theorists assume that emotions change and can be regulated, whereas constant theorists assume that emotions are static and cannot be controlled.

People with incremental implicit theories of emotions (as opposed to constant) tend to prefer productive strategies of emotional self-regulation–cognitive reformulation [45] and cognitive reevaluation [46]. Incremental ITEs are also linked to the suppression of emotions [46]. Interestingly, incremental theories of emotions are important for the adoption of emotional control and suppression strategies, but they are not related to cognitive overestimation [47].

Incremental ITEs are also linked to high effort in terms of emotional self-regulation and too low levels of pathological distress [48], experiencing few negative emotions and additional positive emotions and social support [45], as well as higher proactivity in coping: people with incremental ITEs tend to prefer psychotherapy over medication when seeking help [49].

The study aimed to address the following questions: Which of the two cultures, Russian or Chinese, shows higher scores on measures of empathy and emotional control skills? How are empathy and the preference for ITE correlated in participants from such different countries as Russia and China? We anticipated that the relationships among these variables would differ due to different demands placed on the relevant skills and behaviors in different cultures.

According to numerous studies, men generally show lower empathy compared to women [24,37,50,51,52,53]. These findings are partially supported by neuroimaging evidence suggesting higher levels of activation (indexing effortful processing) in the amygdala during emotional processing, one of the key brain structures involved in subjective emotional reactions and perceptions [54]. What underlies these differences is unknown, given the strong transmission of gender roles as well as gender stereotypes as one of the key and frequent cultural characteristics [54].

Therefore, this study aimed to (1) establish cross-cultural differences in psychological components of empathy and implicit theories of emotion between Russian and Chinese samples; (2) identify latent classes of personality traits using cognitive empathy, affective empathy, and implicit theories of emotion capturing beliefs regarding the possibility of emotional control as indicator variables; and (3) interrogate gender differences in empathy and ITEs in Chinese and Russian participants.

### 1.3. Hypotheses

**Hypotheses** **1.**
*Given the pivotal role of interpersonal relationships in Chinese culture, we hypothesized that Chinese individuals would display higher levels of empathy and implicit incremental theories of emotion than Russians.*


**Hypotheses** **2.**
*Cross-cultural adaptation is strongly mediated by immediate and current influences of the social environment. Thus, we expected that Chinese participants living in Russia would show scores intermediate between Russian participants and Chinese participants.*


**Hypotheses** **3.**
*The adoption of a person-centered approach to characterize individual differences in empathy provides additional information regarding the possible differences in latent personality profiles across the cultural groups compared to the correlational analysis.*


The latter point deserves special consideration. The vast majority of published studies to date examined empathy with respect to its relationships with other traits using correlational designs that could be characterized as variable based. It can be argued that while appropriate for establishing such associations, the method may not be sensitive enough to cross-cultural differences in the pattern of the combinations of traits and subgroups in the study samples. Therefore, we utilized latent class analysis (LCA) to provide an integrative characterization of latent profiles of traits.

As a person-oriented method, LCA has a goal that is similar to the well-known cluster analysis: to establish the separation between X groups in the sample. However, as a model-based method, it provides a strong advantage over cluster analysis because formal hypothesis testing can be performed to identify the optimal cluster structure; importantly, the selection of the optimal cluster structure and the number of clusters can be performed using the formal comparison of model fit indices.

In the current study, we utilized LCA to identify homogenous subgroups of latent classes of individuals who provided similar scores on the study measures, in particular, QCAE. The resolved class structure was then used to compare the profiles in the Chinese and the Russian samples with respect to the interrelationships between empathy and implicit theories of emotion as key components of emotional regulation. Therefore, we hypothesized that the sample would contain several latent classes that differ with respect to QCAE profiles. We also hypothesized that the latent classes should be differentiated by the extent to which they adopt incremental implicit theories of emotions and that cross-cultural differences would manifest as differences in class composition and structure in groups of participants from China and Russia.

## 2. Materials and Methods

### 2.1. Participants

A total of 1322 participants were recruited for the study; the data from 1184 participants were analyzed (see below for exclusions). The Russian sample included 400 of 523 participants who originally took part in the study—adjustments for age and sex differences were performed using propensity score matching to the Chinese living in China group. We removed 16 participants from the sample of the Chinese living in Russia and 2 participants from the Chinese living in China samples due to their age (above 55). As shown in Table 1, the age of Chinese participants in Russia was lower, as most of them were students (92%).

Russians in Russia: 400 people, mostly students (76.5%); 99 of them were men М = 28.47, SD = 8.24) and 301 women (М = 24.07, SD = 6.2).

Chinese in Russia: 363 people, mostly students (92.2%); 247 of them were men (М = 21.45, SD = 6.11) and 115 women (М = 20.74, SD = 3.21).

Chinese in China: 421 people, 56.5% of them were students; 169 of them were men (М = 24.44, SD = 7.10) and 252 women (М = 25.02, SD = 6.89).

All participants were tested on a voluntary basis. The Russian sample consisted of (1) Lomonosov Moscow State University undergraduate, master, and Ph.D. students. They were recruited by their professor and Ph.D. student; (2) online group, recruited according to the principle of “snowball”, where participants were invited by students and two researchers. The Chinese participants consisted of those tested in person and online through an invitation of their fellow students in the Chinese community. Chinese participants received a small compensation (CNY 5–10). All procedures performed in the study were in accordance with the ethical standards of the Ethics Committee of Psychology Faculty from Moscow State University Lomonosov.

### 2.2. Psychological Assessments


Implicit theories of emotion scale (ITE in [45]) includes 4 items, with 2 items indexing constant and incremental theories each. The participants were asked to evaluate the extent to which they agree with a set of statements using a 5-point Likert scale. The outcome metric from this assessment was based on the one-factor solution and indexed the extent to which the person thinks that they can control their own emotions (e.g., by endorsing a statement such as “Everyone can learn to control their emotions”). A preliminary study by our group confirmed a one-factor solution. Using Cronbach’s alpha (internal consistency) coefficient, we estimated the reliability at scale ITE α = 0.623 for Chinese living in China; α = 0.710 for Chinese living in Russia; and α = 0.692 for Russians living in Russia.Questionnaire of Cognitive and Affective Empathy, QCAE [18]; for the Chinese adaptation, see [40,55]. In our adaptation [56], QCAE includes 31 items requiring the participant to indicate the extent to which they agree with the statement using a 4-point Likert scale. Unlike Davis’ IRI [16], QCAE also includes two second-order factors—cognitive empathy and affective empathy. Unlike in other studies [40,55], in our Chinese sample, a 4-factor model of the questionnaire was established. The cognitive empathy component consisted of 2 subscales: Perspective taking (10 items) assesses the extent to which participants can see things from another’s point-of-view (e.g., “I can easily figure out what another person might want to talk about”); online simulation (9 items) measures the extent to which a person tries to or wants to put oneself in another person’s position by imagining what that person is feeling (e.g., “Before criticizing somebody, I try to imagine how I would feel if I was in their place”). The Affective empathy component consisted of 2 subscales: Emotional contagion (8 items) assesses the automatic mirroring of the feelings of others (e.g., “I am happy when I am with a cheerful group and sad when the others are glum”); peripheral responsiveness (4 items) assesses ones’ emotional responsiveness to the moods of others in a detached social context (e.g., “I usually stay emotionally detached when watching a film”).


### 2.3. Statistical Analysis

Correlational analyses, MANOVA, and group comparisons (results in Section 3.1, Section 3.2 and Section 3.3) were conducted using the IBM SPSS software (version 23). Latent class analysis via mixture modeling was conducted using the Muthén & Muthén Mplus software v. 8.6.

Data were dichotomized to arrive at a set of binary indicators for each variable, using a median split. To perform the latent class analysis (LCA), we examined a set of models and such fit indices as Akaike information criterion (AIC) and the Bayesian information criteria (BIC); entropy was indicating the reliability of classification solution; the p-value for the Lo–Mendell–Rubin adjusted likelihood ratio (cut-off at 0.05; LMR) [57], designed to provide formal evidence for preference of K vs. K-1 classes models. Lower AIC and BIC correspond to better fit. Lubke and Muthén suggested that entropy <0.60 would lead to misclassification of 20% of individuals, whereas entropy >0.80 would correspond to classification accuracy over 90%. After selecting the final models, we compared the established latent classes with respect to the indicator variables to further characterize them.

## 3. Results

### 3.1. Group Differences

Using the Kruskall–Wallis test, we found significant differences (*p* < 0.05) between the three groups for all of the study variables (Table 1). Chinese in Russia showed the lowest levels across all variables, while Chinese in China showed the highest levels of cognitive empathy and its constituent subscales (perspective taking, online simulation).

A MANOVA analysis revealed that ITE was related to both the study group and the interaction term between “group * gender”: *p* = 0.0001 and *p* = 0.027, respectively. The levels of cognitive empathy were related to group (*p* = 0.0001) and gender (*p* = 0.006); affective empathy was related to group, gender, and the interaction between them (all *p* = 0.001). Post-hoc follow-up analyses are presented below.

### 3.2. Gender Differences

Gender differences are explicated in a set of Figure 1, Figure 2, Figure 3 and Figure 4. The two Chinese groups were combined for the purpose of this analysis. Using Mann–Whitney’s nonparametric U criterion, we found statistically significant differences between Russian and Chinese men (*p* < 0.01; Figure 1): Russian men were more likely to adopt implicit incremental theories of emotions; Chinese men scored higher on variables of affective and cognitive empathy. We also found a significant (*p* < 0.01; Figure 2) difference between Chinese and Russian women, with Chinese women showing higher levels of cognitive and affective empathy, but similar to Chinese men, they are more prone to the constant implicit theory of emotions (Figure 2).

In the Russian sample, we did not establish statistically significant gender differences in ITE, but we found that cognitive empathy and affective empathy were higher in Russian women than men (both *p*’s < 0.01; Figure 3). In the Chinese sample, all study variables produced statistically significant gender differences, with a consistent profile of higher values in women, compared to men, across them (see Figure 4; *p*’s = 0.001).

Thus, we found similar patterns of gender differences in the two Chinese samples. But we did not combine them into one selection for the following reasons. First, the Chinese in China and the Chinese in Russia group differed with respect to average values on many study variables. Second, we also set out to test Hypothesis 2 about the mediating role of the social conditions of a person living in another culture, which could be reflected in the latent profiles of the samples. The subsequent correlation analysis showed this was justified since the differences between the two Chinese samples in terms of ITE and empathy connections were established.

### 3.3. Correlational Analysis

We conducted a correlational analysis of the relationships between ITEs and empathy in all three groups (see Appendix A) using Spearman’s rank correlation coefficient. We did not establish a significant correlation between ITEs and empathy in Russians and Chinese living in Russia (*p*’s > 0.05). However, in Chinese living in China, we found positive significant relationships between ITEs and cognitive empathy (Spearman’s ρ = 0.212, *p* < 0.001), perspective taking (ρ = 0.150, *p* < 0.01), and online simulation (ρ = 0.238 *p* < 0.001).

### 3.4. Latent Class Analysis

We used a combination of guidance provided by the formalized analysis of model fit indices, as well as prior theoretical considerations to arrive at the final model solution. Therefore, we have settled on the following set of models. In Russians, although AIC, aBIC, and entropy suggested K = 3 classes, K = 2 was a simpler model, consistent with a lack of fit improvement according to the formal LRT test and was the most parsimonious. In Group Chinese in Russia, a two-class model showed better fit than a three-class solution according to AIC, BIC, aBIC, and LRT. In Chinese in China, despite higher levels of entropy in the K = 3 model, AIC, BIC, aBIC, and LRT all pointed to the K = 2 as a better and more parsimonious solution. The fit indices and model comparisons are provided in Table 2.

The global fit indices provide guidelines for the selection of the final model, but it is also necessarily guided by theoretical considerations. Although in the Russian group (N = 400), regardless of the AIC, the entropy criterion favored a three-class solution, and the LRT test did not suggest that a three-class solution provided better fit. Based on simplicity and interpretability, we chose model class 2. For group Chinese in Russia (N = 363), model class 2 demonstrated better AIC, BIC, and aBIC indices; model class 3 demonstrated an LRT *p*-value > 0.05, which means that the model class 3 did not show a better fit than model class 2. Similarly, for the Chinese in Russia group, we chose model class 2 as the final solution. For similar reasons, we chose model class 2 as the final for group Chinese in China (N = 421).Russians (Figure 5)

Latent Class 1 (N = 167) had a low probability of endorsing cognitive as well as emotional empathy items and therefore was labeled as ‘Low empathy’; Class 2 (N = 233) had moderate probabilities of endorsing cognitive empathy and higher probabilities of endorsing emotional or affective empathy and was labeled ‘High empathy’.

Using the Mann–Whitney U criterion to compare the established class differences in study variables, we established significant differences (*p* < 0.05) for all study variables but emotion contagion (*p* = 0.091).Chinese in Russia (Figure 6)

In this group, all study variables produced significant interclass differences (*p* < 0.05). Class 1 (N = 155) was characterized by a high probability of endorsing most study items but those related to peripheral sensitivity and was labeled as ‘High empathy’; Class 2 (N = 208) showed the opposite pattern and was labeled ‘Low empathy’.Chinese in China (Figure 7)

In this group, we established significant (*p* < 0.05) differences between the two classes across all variables but peripheral sensitivity (*p* = 0.205). As shown in Figure 7, Class 1 (N = 216) endorsed most items but peripheral sensitivity and was labeled ‘High empathy’. Class 2 showed the opposite pattern (N = 205) and was labeled ‘Low empathy’.

The results of LCA suggest a similar latent structure in Chinese living in Russia and Chinese living in China, with class separation mostly conveyed by cognitive empathy. It is worth noting that in this sample, higher cognitive empathy was associated with lower peripheral sensitivity. However, this was not the case for Russians (Figure 5), for whom the two classes were mostly differentiated by affective or emotional empathy.

### 3.5. Class-Specific Correlational Analysis

In Russians and Chinese living in Russia, correlational analysis (using Spearman’s rank coefficient) did not reveal significant relationships between ITEs and empathy in either of the latent classes. In Chinese living in China, we found that in both classes, ITE was positively correlated with cognitive empathy: for “high empathy” class (ρ = 0.210, *p* = 0.002) and for “low empathy” class (ρ = 0.158, *p* = 0.0025). We also found positive correlations between ITEs and Online Simulation for both class 1 (ρ = 0.255, *p* = 0.000) and class 2 (ρ = 0.140, *p* < 0.05).

### 3.6. Relationships in Student Subsamples

When restricting the analyses to subgroups of students across the study groups (Russians, Chinese in Russia, and Chinese in China), significantly higher cognitive and affective empathy levels were replicated among the Chinese living in China. As in the general sample of Chinese in China, an increase in the incremental ITE indicator was associated with cognitive (ρ = 0.237, *p* = 0.01) and affective empathy (ρ = 0.141, *p* = 0.05). Although cognitive empathy and affective empathy were positively associated in all three student samples, this particular relationship was only seen in Chinese in China.

A sensitivity analysis of the LCA solution identified the same two classes across three groups.

In Russian students in the “high-empathy” class, in contrast to the general Russian sample, a positive relationship was established between the increased preference for incremental theories of emotions and cognitive empathy (ρ = 0.199, *p* = 0.01). Still, in the “low-empathy” class, no relationship has been established. In a subsample of Chinese students in Russia in the class of “high-empathy”, a positive relationship was also found between an increase in preference for incremental theories of emotion and cognitive (ρ = 0.181, *p* = 0.05).

Finally, in a subsample of Chinese students in China, a positive correlation of incremental ITEs with cognitive empathy was identified: ρ = 0.207, *p* = 0.05 for the “high-empathy” class, and ρ = 0.219, *p* = 0.005 for the “low-empathy” class. In addition, a positive relationship was established between an increase in incremental ITEs and affective empathy (ρ = 0.258, *p* = 0.01) in the “low-empathy” class.

## 4. Discussion

Our cross-cultural study of empathy established, in line with our initial predictions, that empathy was notably higher in a more collectivistic China in comparison with a more individualistic Russia. However, regarding the ITE, we accept the counterhypothesis—incremental theories of emotions were more pronounced and prevalent among Russians than the Chinese. We could assume that the Chinese culture particularly values cognitive control of emotions. However, Russians scored significantly higher on this scale as well. At the same time, beliefs about managing emotions in the Russian sample are not related to the assessed levels of empathy, suggesting that beliefs regarding one’s ability to control emotions are not integrated into the processes of empathy. Similar results of the comparatively autonomous functioning of empathy as a variable in the emotional-personal sphere and ITE as a manifestation of the mindset turned out to be characteristic of the Chinese living in Russia. Intriguingly, the two Chinese groups demonstrated separation by peripheral sensitivity, which was surprisingly lower in the “high-empathy” classes, suggesting that it indicates compensation in relation to other empathic components.

The social conditions of temporary living and adaptation of the Chinese in a different country appear to partially regulate the levels of empathy: it is lower among the Chinese living in Russia (in Moscow) and is similar to the levels found in Russians. The Chinese in Russia demonstrate a decrease in perceived emotional control, including those characterized by higher affective empathy, which may indicate a greater malleability of the personality’s self-consciousness to current sociocultural influences. Thus, we accept the first two hypotheses of the study regarding empathy. Empathy was higher among the Chinese in China. Differences in sociocultural conditions (living in China or Russia) affect the integration of ITE with empathy in general emotional regulation. In this respect, the sample of Chinese in Russia was similar to the sample of Russians.

Previous research showed that empathy is related to the tendency to orient towards one’s inner group. Our study suggests that empathy is dynamic and susceptible to strong contextual influences that are juxtaposed onto cultural differences. These effects likely have a multifactorial architecture with bidirectional links: The observed decrease in empathy in Chinese living in Russia simultaneously captures the environmental factor of continuous engagement with their familiar, culturally united society and adaptation to being abroad in a foreign country culture.

This finding has important practical implications for our understanding of the adaptation process to foreign cultures, especially considering the large and growing inflow of Chinese students into Russian universities, which highlights the need for evidence-based academic and sociopsychological adjustment support services. It concerns the change in personality traits and, as demonstrated via latent class analysis, the change in the relationship between empathy and implicit theories about the controllability of emotions. Within the latent classes established for the Chinese in China, a lower incremental ITE was linked with cognitive empathy. In contrast, the relationship between empathy and ITE was not evident in the Russian sample classes.

Thus, our class-specific correlational analysis demonstrated that for Russians and Chinese living in China, the observed relationships were similar, while Chinese living in Russia showed a somewhat unique pattern. This is consistent with the general framework that suggests partial dysregulation of emotional regulation, particularly empathy in Chinese living abroad. Thus, higher empathy in this case can correspond to a diminished orientation towards others and enhanced beliefs about one’s inability to change emotions.

It should be noted that the student subgroups in the Russian and Chinese samples, on the one hand, replicated the general patterns of the results in terms of the identification of latent classes. On the other hand, the established relationships between incremental ITE and cognitive empathy precisely in high-empathy classes (and their absence in low-empathy classes) reinforce the redirection of effective empathy towards cognitive processes in student subsamples of Russians and Chinese in Russia. The connection between ITE and affective empathy in the “low-empathic” class of Chinese students in China can also be interpreted similarly: high empathy is achieved via the development of cognitive empathy rather than affective empathy. This inference is relatively consistent with the idea of understanding empathy modeling as a higher mental function [23].

Our analyses support the third hypothesis that Latent Class Analysis provides new and incremental information compared to correlational analysis and analyses of group differences in the average levels of individual indicator variable. In particular, the content analyses demonstrate a higher integration of emotional-personal domains by the Chinese in comparison to the Russians.

Some recent studies (e.g., [58]) did not establish significant gender differences in empathy in Chinese samples. These results may be attributable to the use of a particular assessment, the Jefferson Scale of Empathy (JSE), which was developed and targeted for use with medical students. Therefore, the scale may have biased the domain representation of empathy, especially given the widely acknowledged differences in empathy between members of different occupations, with high overall empathy in medical doctors. However, in our study, the established pattern of gender differences is concordant with the profile of results from many other studies [20,50,51,52]. We found that in the Chinese sample, women exhibited higher levels of affective and cognitive empathy, as well as the adoption of implicit incremental theories of emotions, rooted in a belief that emotions can be modified and controlled than men. On the other hand, Russian women only showed higher levels of cognitive and affective empathy than men, suggesting a much more moderate profile of gender differences, consistent with the value systems of Western cultures.

Implicit theories, therefore, index beliefs regarding one’s ability to regulate one’s emotions, which in turn has serious implications for real-world behavior in situations that elicit emotional reactions (replete in everyday life). Chinese culture pays close attention to relationships among objects and relationships between objects and the environment [59]. We demonstrated that this holistic framework underlies the more integrated relationship between implicit theories of emotion and cognitive empathy, established uniquely for the Chinese.

## 5. Limitations

Note that among the Chinese in China, ITE was associated primarily with cognitive empathy. At the same time, a more cognitively mediated control of empathy in the Chinese samples was accompanied by the greater adoption of constant implicit theories of emotions compared to the Russian sample. In this, we see some contradictions, which can be explored in more detail in subsequent studies.

Another limitation is related to the fact that our sample of Chinese in Russia is mainly represented by students, who may be characterized by specific patterns of personality traits or intellectual abilities that allow them to choose to proceed with an abrupt change in lifestyle, moving to another country to study and facing a number of uncertainties and challenges, thereby leading to self-selection. Including the dimensions of empathy in a broader context of relationships with other variables of intellectual-personal potential can help study the relationship between affective and cognitive processes that mediate empathy as a result.

Although cross-cultural studies of cognitive and emotional components of empathy are uniquely positioned to advance cross-cultural communication and facilitate it, the study only examined proxies for variables that could be predictive with respect to actual decision making under uncertainty and risk in real-life situations. Future studies should therefore focus on various cross-cultural manifestations of the relationship between empathy and emotional regulation of moral choice, as well as other emotion-colored decision-making situations.

Finally, certain limitations are inherent to the use of self-reported data that may or may not adequately represent process components of empathy as a state and ability. Interestingly, women reported higher empathy when measured by questionnaires [24,50,52] but not when physiological measures were used [60]. The field is ready for further methodological developments that adequately capture the distinction between trait and state empathy, as also noted by an anonymous reviewer of this manuscript.

## 6. Conclusions


We established significant cross-cultural differences between the Chinese and the Russian groups: individuals from China showed higher cognitive and affective empathy levels and tended to adopt more constant implicit theories of emotion, compared to Russians. Chinese living in Russia showed lower empathy compared with Chinese living in China. Russians demonstrated the highest levels of adoption of incremental theories of emotion, including emotional control.The role of current and immediate social surroundings is evident in a preferential coupling between incremental implicit theories of emotion and empathy in Chinese living in China but not Chinese living in Russia. These findings suggest a dissonance in emotional regulation in those individuals who live abroad, manifested in this case as negative relationships between empathy and measured orientation towards the emotions of others as well as belief in malleability and controllability of one’s emotions.Latent class analysis is a powerful person-centered technique that identified two distinct latent classes of participants that mapped onto several types of psychological regulation that differentiated between different cultural groups.The gender differences are less pronounced in the Russian sample in comparison to the Chinese sample, which can be interpreted as evidence towards viewing Russia as being more aligned with the individualistic West and China as more collective.


## Figures and Tables

**Figure 1 behavsci-11-00137-f001:**
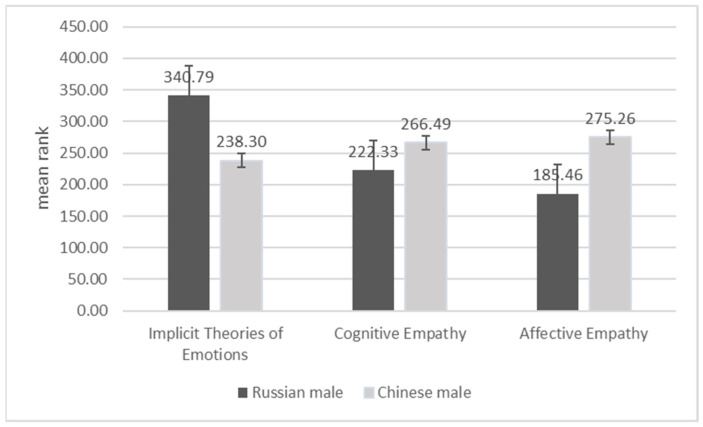
Differences between Russian and Chinese men on implicit theories of emotions, cognitive empathy, and affective empathy.

**Figure 2 behavsci-11-00137-f002:**
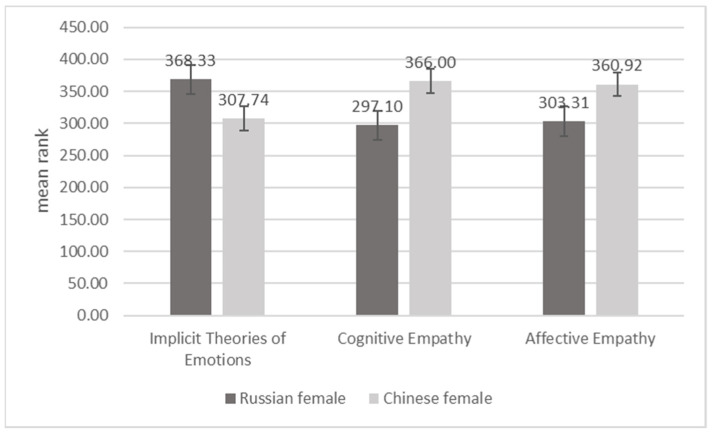
Differences between Russian and Chinese women on implicit theories of emotions, cognitive empathy, and affective empathy.

**Figure 3 behavsci-11-00137-f003:**
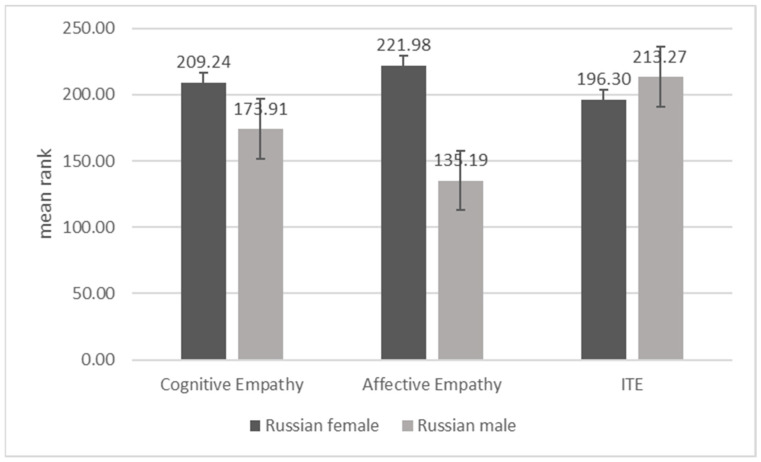
Difference between Russian females and males on cognitive empathy and affective empathy.

**Figure 4 behavsci-11-00137-f004:**
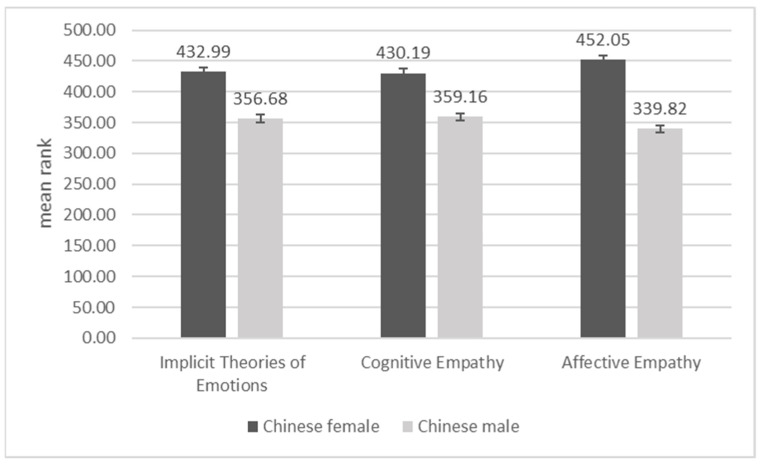
Differences between Chinese females and males on implicit theories of emotions, cognitive empathy, and affective empathy.

**Figure 5 behavsci-11-00137-f005:**
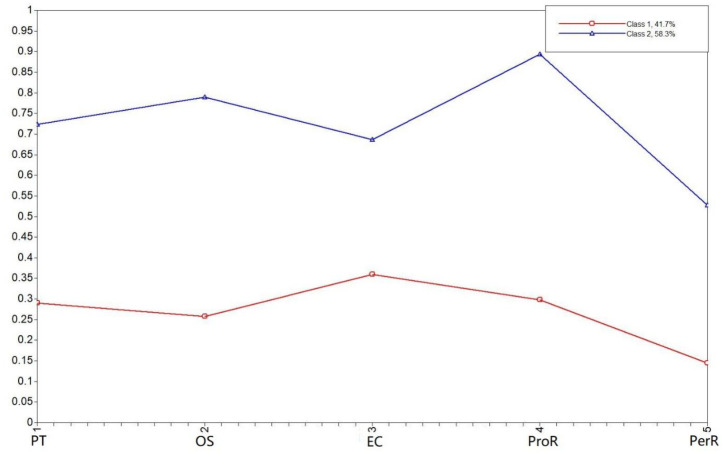
Class-specific response probabilities for each of five subscales of Empathy in Russian Group. Note: PT means Perspective Taking; OS—Online simulation; EC—Emotion Contagion; ProR—Proximal Responsivity; PerR—Peripheral Responsivity.

**Figure 6 behavsci-11-00137-f006:**
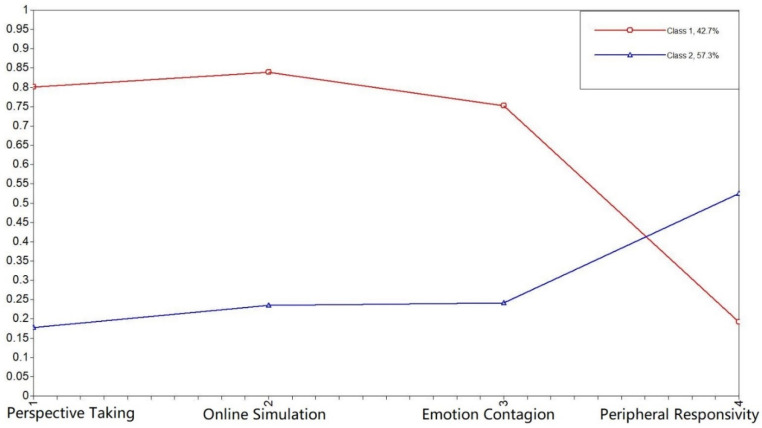
Plot of class-specific response probabilities for each of four subscales of Empathy for Chinese in Russia. Note: our previous work showed that a four-factor structure performs better for the Chinese sample.

**Figure 7 behavsci-11-00137-f007:**
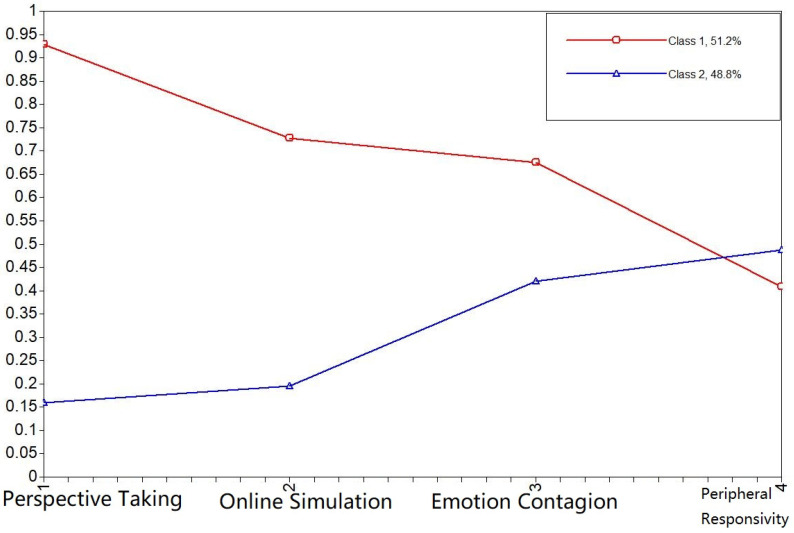
Plot of class-specific response probabilities for each of four subscales of Empathy for the Chinese in China.

**Table 1 behavsci-11-00137-t001:** Descriptive statistics for study variables.

	Russians in Russia		Chinese in Russia		Chinese in China			
Variable	M	SD	M	SD	M	SD	X^2^	*p*
1. Implicit Theory of Emotions	3.60	0.84	3.10	0.41	3.43	0.72	117.20	0
2. Perspective Taking	28.76	4.20	27.05	4.16	28.63	4.77	47.69	0
3. Online Simulation	22.94	3.27	23.84	3.43	27.36	3.70	298.24	0
4. Peripheral Responsiveness	9.06	1.91	10.06	1.55	10.26	2.21	77.70	0
5. Cognitive Empathy	51.69	6.23	50.89	6.81	55.99	7.59	129.32	0
6. Affective Empathy	30.66	4.65	30.40	3.71	32.86	4.88	76.48	0
7. Age	25.16	7.02	21.32	5.68	24.78	6.97	132.65	0

Note. Group 1 (Russians): N = 400; Group 2 (Chinese in Russia): N = 363; Group 3 (Chinese in China): N = 421.

**Table 2 behavsci-11-00137-t002:** Comparison of model fit indices for alternative LCA solutions in three groups.

Model	AIC	BIC	aBIC	Entropy	aLRT P
Russian group, class(1)	2705.82	2725.78	2709.91	/	/
**Russian group, class(2)**	**2574.12**	**2618.03**	**2583.12**	**0.59**	**0**
Russian group, class(3)	2571.60	2639.46	2585.52	0.66	0.34
Russian group, class(4)	2575.07	2666.88	2593.90	0.83	0.13
Chinese in Russia, class(1)	1993.78	2009.36	1996.67	/	/
**Chinese in Russia, class(2)**	**1868.29**	**1903.34**	**1874.79**	**0.60**	**0**
Chinese in Russia, class(3)	1871.13	1925.65	1881.23	0.60	0.14
Chinese in Russia, class(4)	1878.28	1952.27	1891.99	0.75	0.37
Chinese in China, class(1)	2326.75	2342.92	2330.23	/	/
**Chinese in China, class(2)**	**2242.10**	**2278.49**	**2249.93**	**0.60**	**0.0000**
Chinese in China, class(3)	2246.75	2303.35	2258.92	0.65	0.19
Chinese in China, class(4)	2254.18	2330.99	2270.70	0.59	0.45

Note. Russian group: N = 400; Chinese in Russia: N = 363; Chinese in China: N = 421.

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
