# Peer review of "Cross-Cultural Comparison of Relationships between Empathy and Implicit Theories of Emotions (in Chinese and Russians)"

_behavsci, 2021, doi:10.3390/bs11100137_

Round 1

Reviewer 1 Report

Title: It needs some changes: implicit theories should be replaced with implicit theories of emotion.

Abstract: It is proposed to highlight the theoretical framework of the study and to present the research question.

Introduction: It is proposed to strengthen the theoretical introduction in order to clarify the research question together with the proposed hypotheses. What is missed in the literature and what would be improved in the proposed study?

Materials and Methods: The sample should be described more carefully:

1) proportion of males and females in each group,

2) sociodemographic characteristics (occupation, level of education etc.).,

3) strong arguments about the equivalence of three compared groups.

4) number of participants mentioned in Abstract part (p.1) is different from the one in Materials and Method part ( p.5);

Procedure: It is not clear how the participants were recruited for the study. It is proposed to explicit the procedure of the study.

Discussion: It is proposed to deepen this part of the manuscript in order to highlight the contribution of the results of the presented study into the scientific field.

Appendix: Some words are in Russian. It is proposed to use the same style to present the results in the Table.

In general, more attention is paid to the Chinese culture than to the Russian one (it is clear from the text itself and the list of references), however, the title makes think that these two cultural groups are equal.

Author Response

Point 1 : Title: It needs some changes: implicit theories should be replaced with implicit theories of emotion.

Response 1 : We agree with the reviewer, and have modified the title of the manuscript accordingly.

Point 2: Abstract: It is proposed to highlight the theoretical framework of the study and to present the research question.

Response 2: We have restructured the abstract to include (sentences 2 through 4) the key research question while highlighting, within the review, the relationship between cross-cultural factors and the profile of personality and emotion traits in Russians and Chinese.

Point 3 : Introduction: It is proposed to strengthen the theoretical introduction in order to clarify the research question together with the proposed hypotheses. What is missed in the literature and what would be improved in the proposed study?

Response 3 : Thank you for this suggestion. We have revised the Introduction and restructured it, leading to the description of the design of the study. We have also included a separate section on implicit theories of emotion.

We now clarify that the published literature to date has largely ignored cross-cultural differences in the emotional domain of personality when comparing Western and Eastern cultures. Importantly, we also note that implicit theories of emotion have also not been included in consideration with respect to their relationships with empathy. Finally, to the best of our knowledge, there are currently no published empirical reports employing the person-centered approach that would contextualize cross-cultural differences in terms of the presence of latent homogenous subgroups of individuals characterized by specific patterns of absolute values on as well as relationships among indicator variables. 

Point 4 proportion of males and females in each group,

Response 4: We have modified the Methods to include the requested information.

Point 5: sociodemographic characteristics (occupation, level of education etc.).,

Response 5: Although we did not collect detailed data on socio-economic status and specific occupation, all participants reported whether they were current students vs. held a job.

Point 6: strong arguments about the equivalence of three compared groups.

Response 6: The study recruitment followed a hybrid approach and included a snow-ball component, particularly within the student populations. We report the statistics on socio-demographics for all study groups. In addition, to improve the equivalence of the groups, we have performed propensity score matching, individually matching participants to Chinese living in China from the other two groups, which led to a minor decrease in sample size but improved age distributions and gender ratios.

Point 7 : number of participants mentioned in Abstract part (p.1) is different from the one in Materials and Method part ( p.5);

Response 7 : Thank you for catching this! We have revised the numbers throughout.

Point 8: Procedure: It is not clear how the participants were recruited for the study. It is proposed to explicit the procedure of the study.

Response 8: Thank you, we now provide a brief description of the study protocol. Briefly, we note that in Russia participants were recruited via Moscow State University or via the Internet flyer advertising the study. We did not compensate the participants. The Chinese participants consisted of those tested in person and online through an invitation of their fellow students in the Chinese community.

Point 9: Discussion: It is proposed to deepen this part of the manuscript in order to highlight the contribution of the results of the presented study into the scientific field.

Response 9: We completely agree with the reviewer, and have modified the text accordingly. Discussion has also been broadened to reflect that.

Point 10: Appendix: Some words are in Russian. It is proposed to use the same style to present the results in the Table.

 Response 10: Thank you, we have fixed this issue.

Point 11: In general, more attention is paid to the Chinese culture than to the Russian one (it is clear from the text itself and the list of references), however, the title makes think that these two cultural groups are equal.

Response 11: We now provide a description of the two examined cultures from the standpoint of Hofstede’s individualism-collectivism continuum, and highlighted both Russia’s and China’s shifting positions on it. We hope that the title reflects this juxtaposition by referring to both universal (common) and unique (non-shared) cultural elements, which is pivotal for all, if not most cross-cultural studies. We thank the reviewer for the detailed and thoughtful suggestions, and hope that we improved the manuscript sufficiently.

Reviewer 2 Report

This study is conceptually interesting and more cross-cultural research in the area of empathy and emotion is certainly warranted. Unfortunately, in the form presented here, I do not think that the manuscript is suitable for publication in the journal.

  • The introduction seems disproportionally long and lacks a clear focus. Amongst many (superfluous) details it is difficult to pick out what specific aspects motivated the study, what literature the hypotheses that were tested are based on, and why this specific study should be considered relevant.
  • The methods section does not give any information about how the data were obtained (For example, how were study participants recruited? Were the data collected online or paper-based? Was there an institutional ethics votum?).
  • Similarly, the methods section does not provide much detail about the actual analysis procedure (Were the data cleaned prior to analysis? How many data points were missing/discarded? Is the data publicly available? What software and/or packages were used for the analysis? What reference or analysis protocol were the different analytic steps based on? Is the coding script publicly available?).
  • The three groups differ considerably in terms of mean age, which potentially confounds all between-group comparisons that are reported. Unfortunately, this seems not to be considered in either the analysis or (at least) the discussion section. Similarly, when comparing differences between male and female participants, potentially confounding factors (such as age) should also be taken into consideration for the analysis.
  • It would be relevant to know how long participants in Group 2 (Chinese participants living in Russia) were living in Russia, otherwise it is difficult to assess the plausibility of the claim that “living abroad is a significant modifier of the latent structure of emergent empathic states”.
  • There is no mention of pre-registration (or at least an a-priori specification) of hypotheses, so especially the correlation results are difficult to interpret (correlating groups on multiple variables without correcting for multiple comparisons will easily yield false positive results). If there was no a-priori formulation of pre-registration of the hypotheses, this should at least be acknowledged as a limitation.
  • There is a number of grammar and spelling mistakes (e.g. “...cross-cultural studies that of its role...”, “During one the previous epidemic...”, “during the COVID-19”, “...men were hypothesized that men’s thinking...” etc.)

Author Response

  • Point 1 : this study is conceptually interesting and more cross-cultural research in the area of empathy and emotion is certainly warranted. Unfortunately, in the form presented here, I do not think that the manuscript is suitable for publication in the journal.

Response 1 : Thank you for recognizing the value of the study as well as the examined problem. We agree that the initial version of the manuscript might have lacked some clarify and structure, in particular with respect to contextualizing the study. We have revised the manuscript while clarifying the theoretical foundations for the study. Detailed responses to individual feedback points are presented below.

  • Point 2: The introduction seems disproportionally long and lacks a clear focus. Amongst many (superfluous) details it is difficult to pick out what specific aspects motivated the study, what literature the hypotheses that were tested are based on, and why this specific study should be considered relevant.

Response 2: We have restructured the theoretical part of the manuscript. We now explicate that the study has been motivated by the lack of published data on emotion and associated traits with respect to cross-cultural differences; virtual absence of data on the relationships between beliefs about emotions (i.e., implicit theories of emotion) and empathy; a general bias towards variable-centered approaches that neglect sample substructures as valuable sources of information about the traits, and, most importantly in the context of a cross-cultural study, groups of individuals that belong to different cultures. Thus, our study is one of the first efforts to characterize the relationships among the emotional components of the personality domain. Importantly, by recruiting a group of Chinese living in Russia, we also enabled the inference about the moderating role of the immediate social context/environment on the observed cross-cultural differences.

Finally, we have revised the Discussion to highlight the practical importance of the study findings with respect to tracking changes in the emotional domain in vulnerable populations – e.g., students living abroad in a substantially different cultural context. Furthermore, we note that the particular comparisons performed within the study are particularly noteworthy – while Russia is considered a Western culture, relative to China, it is viewed as more Eastern culture in Europe and the U.S. The study therefore also  aimed to bridge a gap in our understanding of the key difference between Russian and more traditional Easter cultures. We hope that our revisions have made the manuscript more streamlined while addressing the reviewer’s concerns and providing a much richer contextualization for the problem than a (frequently) simplified framework for understanding cross-cultural differences in emotion between the West and the East.

  • Point 3: The methods section does not give any information about how the data were obtained (For example, how were study participants recruited? Were the data collected online or paper-based? Was there an institutional ethics votum?).

Response 3: We have revised the Methods section to include the recruitment information. Recruitment was performed using a hybrid approach – “snow-balling” via word of mouth and via a combination of advertisements. Acknowledging (small but statistically significant) differences in gender ratios and age distributions between the recruited samples, we have attempted to at least partially mitigate that by using propensity score matching and selecting samples that were closest by gender and age distribution to the group of Chinese living in China (e.g., the Russian sample was reduced to N=400 to enable age and gender distribution comparability). We now also provide information on how many participants filled out the assessments in a paper-and-pencil format, and how many – via online forms. Participants were recruited on a voluntary basis. Attached to this response is also a note from the Internal Review Board / Ethics Committee at the Department of Psychology at Lomonosov Moscow State University, where the study was initiated and conducted.

  • Point 4: Similarly, the methods section does not provide much detail about the actual analysis procedure (Were the data cleaned prior to analysis? How many data points were missing/discarded? Is the data publicly available? What software and/or packages were used for the analysis? What reference or analysis protocol were the different analytic steps based on? Is the coding script publicly available?).

Response 4:We now provide additional information of data cleaning (e.g., matching) and transformation, where applicable (e.g., for the LCA). We have recruited additional participants from Moscow State University, extending it to N=523, since the initial submission of the manuscript (prior to matching). We also note that we excluded individuals over the age of 55 (N=15 from China).

Collected data (available from data upon reasonable request) were entered or exported from the Google Forms into an Excel database. We now list the software packages used for statistical analysis and their versions in Methods. For Mplus, we can provide, upon request, the script that was used to perform mixture modeling/LCA, and the steps carried out to perform this analysis are presented in detail in the manuscript. The SPSS-based analyses were conducted via GUI. Please let us know if more information is required.

  • Point 5: The three groups differ considerably in terms of mean age, which potentially confounds all between-group comparisons that are reported. Unfortunately, this seems not to be considered in either the analysis or (at least) the discussion section. Similarly, when comparing differences between male and female participants, potentially confounding factors (such as age) should also be taken into consideration for the analysis.

Response 5: Please see the other responses above. We have now performed additional recruitment/testing as well as propensity score matching to ameliorate the noted differences, which was not fully possible, in part due to the high statistical power for detecting small differences at this N (we provide a detailed description of the composition of the sample to the extent that the data was available to us). The analyses of gender differences, repeated using a set of ANCOVAs, did not detect effects that changed when age was entered as a covariate 

  • Point 6: It would be relevant to know how long participants in Group 2 (Chinese participants living in Russia) were living in Russia, otherwise it is difficult to assess the plausibility of the claim that “living abroad is a significant modifier of the latent structure of emergent empathic states”.

Response 6: On average, this time was estimated at 2-3 years after arrival into Russia. This period is sufficient for substantial changes driven by adaptation related to cultural immersion, intense academic work, and learning a new language. The study validity is enabled by fairly high N but also the utilization of measurement instruments validated for both Russian and Chinese samples. The testing was always performed in a participant’s native language and by a test administrator from their culture, mitigating the concerns related to cross-cultural difference stemming from linguistic and/or translation difficulties.

  • Point 7: There is no mention of pre-registration (or at least an a-priori specification) of hypotheses, so especially the correlation results are difficult to interpret (correlating groups on multiple variables without correcting for multiple comparisons will easily yield false positive results). If there was no a-priori formulation of pre-registration of the hypotheses, this should at least be acknowledged as a limitation.

Response 7: We partially agree with the reviewer that the initial manuscript lacked focus, and have adjusted the text accordingly, rooting it a number of specified hypotheses (these same hypotheses contextualize study findings in the Discussion). We note that the formulation of these hypotheses and study aims necessarily is rooted in what the field already knows. Our study adds to the literature by formulating several directional hypotheses while leaving room for more general statements. We also note that one of the main aims of the study was to detect latent classes of participants. These, by default, do not contain a priori ‘false’ relationships, as hypothesis testing is actually not performed on individual (e.g., pairwise) correlation coefficients. We refer the reviewer to the relevant literature on mixture modeling and its assumptions that in part mitigate the very issue of multiple comparisons at individual variable level.

  • Point 8 There is a number of grammar and spelling mistakes (e.g. “...cross-cultural studies that of its role...”, “During one the previous epidemic...”, “during the COVID-19”, “...men were hypothesized that men’s thinking...” etc.)

Response 8: We would like to thank the reviewer for the thoughtful feedback. We have substantially revised the manuscript, and have enlisted the help of a native English speaker who is a published researcher. Their feedback has been incorporated, and we hope that this has addressed the concerns the reviewer had.

Reviewer 3 Report

Dear colleagues, I hope this message find you well.

Thank you for giving me the opportunity of reading the work “Cross-cultural comparison of relationships between empathy and implicit theories (in Chinese and Russians), it has been a very big pleasure to collaborate reviewing this manuscript. The topic of this paper is very interesting and it seems necessary to delve it. However, there are several questions to improve before to publish it. I would suggest some changes: 

Title and Abstract:

  • Ok

Introduction:

  • The structure of the introduction is not clear. I recommended to divide the introduction into several subsections. For example, I recommend to create a specific subsection where describe each variable and to finish this part describing aims specifically.

Method

  • Study population was poorly described (age, gender, social status…)
  • Why did you choose Chinese and Russian people? A better explanation is required.
  • Data Collection: I recommend adding sample items to each questionnaire.

Results

  • Ok

Discussion

  • It is necessary to describe in more detail the practical and theoretical implications of this research.

Author Response

Point 1 Title and Abstract:

  • Ok

Response 1: We thank the reviewer for their kind words, and for the appreciation of the effort, as well the useful suggestions for improving the manuscript. We have revised the manuscript extensively, and hope that it addresses the reviewer’s concerns. Please see below for specific responses.

Introduction:

  • Point 2: The structure of the introduction is not clear. I recommended to divide the introduction into several subsections. For example, I recommend to create a specific subsection where describe each variable and to finish this part describing aims specifically.

Response 2: Thank you, we have restructured the Introduction section of the manuscript, and added subheadings (mapped onto the key study variables), where applicable; we now also provide a detailed transition to the study design.

Method

  • Point 3: Study population was poorly described (age, gender, social status…)

Response 3: We agree that we have previously provided insufficient detail. We revised the Method section and added information on the recruitment and composition of study groups.

  • Point 4: Why did you choose Chinese and Russian people? A better explanation is required.

Response 4: The study is motivated, first, by the ongoing programmatic research at our laboratory at Moscow State University, conducted by the authors. Second, the study presents the results of a doctoral dissertation study conducted by Ms. Zhou, and at least partially reflects the access to populations available to the authors. Finally, studying emotion in the context of personality traits requires determining whether empathy and beliefs about emotions are affected by culture. Russian and Chinese cultures are strongly separated on the individualism-collectivism continuum, and yet both have unique features that differentiate them from the vastly more studied (and largely) English-speaking Western cultures. We note that these differences are far from being clear-cut, and yet, we formulate a set of theory- and evidence-based hypotheses in the beginning of the manuscript, to aid the reader.

  • Point 5: Data Collection: I recommend adding sample items to each questionnaire.

Response 5: We provide references to the questionnaires that are in public domain and/or available from their respective authors upon request. We now provide more detailed data on the used questionnaires and their psychometric properties in the manuscript.

Discussion

  • Point 6: It is necessary to describe in more detail the practical and theoretical implications of this research.

Response 6: We agree with the reviewer and thank them for their suggestion. We have revised the manuscript to include additional language related to the implications of the study. The practical value of the study lies in its ability to inform culturally-sensitive interventions that can aim to aid in cross-cultural adaptation. From the theoretical standpoint, we argue that the results of the study demonstrate the value of moving towards viewing emotional domain as integrative and including lower-Consciousness (e.g., implicit theories) concepts as well as empathy, a dimensional construct. We hope that the manuscript is improved in accordance with the reviewer’s recommendations.

Reviewer 4 Report

I am reviewing “Cross-cultural Comparison of Relationships between Empathy and Implicit Theories (in Chinese and Russians)” for Behavioral Sciences.  I believe that the authors are trying to examine the factors that predict empathy across three samples: Russians, Chinese, and Chinese living in Russia.  However, the reliabilities for the ITE are very low, the samples are different sizes and ages, and the patterns that the authors say are similar in the figures do not look similar at all.  The authors should also evaluate and display the many dependent variables across gender and cultural group in a table or figure.  I will make suggestions that may help the authors simplify their analyses to evaluate whether the different samples show similar relations between predictors of empathy and empathy.  The prose of the paper suffers from the lack of clarity in many places, and I will provide several examples for the authors that they can use to fix the English throughout the paper.

The ITE reliabilities are woefully low.  The authors could re-run the analyses for the entire sample, created from combining all three samples, and remove items that lower the alpha levels below .7 for the ITE and any other measures.  The reliabilities should be provided for every measure in the study.  The authors should evaluate measures with reliabilities at or above .7 across gender and sample type (3) and they should statistically control for age.  The authors should also statistically control for the sample size if possible.  The authors should examine the measures across gender and sample type controlling for age and put that information in tables or figures.

Rather than conducting several path model analyses across the three samples, the authors should consider conducting regression analyses across the three samples with empathy as the criterion and the same variables as predictors.  The same predictors should be placed in hierarchical regression equations in the same order based on the literature.  In fact, the authors should rework the introduction to set up the order of those variables regardless of the analysis type.  The authors should statistically control for age (put first in the regression analysis) and they should control for sample size if possible.  The current modeling approach is fine if the authors can control for sample age and, hopefully, sample size.  I think that sample size is difficult to control statistically, but the results may be different or similar simply because the samples differ so dramatically in size.  The authors could randomly pull participants from the larger samples to make sure that all three samples are similar in size; I do not like this approach if sample size can be controlled statistically.    

The current analyses with median splits and models with multiple factors that change across the three samples is super confusing.  In addition, the figures that the authors display and refer to as similar do not look similar at all.  Instead, the authors can present the Beta weights for the factors in the regression equation and a comparison of those Beta weights can be made and conclusions can be drawn.

The English is very confusing in several spots in the paper.  I will start with the abstract. Lines 14-17 should say “differences. Chinese living in China showed higher cognitive and affective empathy levels than Chinese living in Russia, and affective empathy was higher overall in women than in men. Finally, women in the combined Chinese sample showed higher affective and cognitive empathy levels as well as incremental implicit theories than men in the combined Chinese sample.”  Was this last comparison actually made?

Line 25 should put commas around “therefore” and line 27 should put a period after “collective” and start the new sentence with “Collectivistic cultures”.  Line 30 should start “Importantly, cultural values are susceptible”.  Line 35 should put a period after “self-construal” and capitalize “Interdependence” for the start of the next sentence.  Line 37 should say “social relationships, with a realization”.  Lines 39-43 should say “Wang and Conway [35] asked participants to recall 20 events and they found the American participants provided more….than the Chinese participants, whereas the Chinese participants were likely to recall…and they detailed social interactions.”

Lines 44-45 should say “for people sharing occupied territory and language as well as a rich set of…”  The next sentence should start “For example, Western and Eastern”.  Line 49 should say “differ in the way individuals”.  Line 59 should say “and it is an important….determining social distancing…”.  Line 63 should say “During a previous epidemic”.  Line 64 should say “but they also reported more”.  I have no idea what lines 65 and 66 are saying to the point that I cannot recommend help.  Lines 67-68 should say “behavior is influenced substantially by culture, which particularly includes empathy and implicit theories…”.  Line 71 should say “multifaceted state and trait”.  Line 72 should delete the text after reference 17 in that sentence.  Line 74 should say “whereas other individuals”.  Line 75 should say “shapes, ranging from emotional”.  Line 78 should put the e.g., information in parentheses.  I have no idea what is being said n lines 80-81.  Line 85 should say “those individuals who score high”.  I have no idea what is being said in lines 84-87. 

I believe the “&” on line 93 should be “and”.  Line 94 should say “studied coping strategies for women experiencing normal pregnancies in Japan and America.”  Line 96 should say “predictor for decisions to social…Russian participants but not in Chinese participants”.  Line 98 should say “students who either”.  Line 99 should say “China. By measuring a wide”.  I have no  idea what is being said in lines 99-100 and in line 105.  I have no idea what is being said in lines 109-113.  Line 114 is confusing.  Line 115 should say “with the following Big Five traits: Conscientiousness, Agreeableness, and Openness”.  Line 119 should say “understand other individuals to explain”.

Line 137 should say “ Implicit…are a domain of individual differences….that is studied less frequently than the domain of empathy or…”  Line 143 should put parentheses around the e.g., information.  Line 151 cannot start with an acronym.  Line 151 says “those of ability” and this phrasing is very confusing. Line 153 should say “whereas constant theorists assume”.  Line 156 should say “but they are not related”. Lines 158 and 159 should put a period after the 55 reference and capitalize “Yet” to start a new sentence.  Lines 160-162 should say “knowledge, no research study has directly examined the pivotal….regulation, and its relationship to empathy.”

Line 163 should say “domain than …is the domain of gender differences”.  Line 163-164 should say “work, men’s thinking was characterized”.  Line 165 should say “outside world, whereas women’s thinking, in contrast, was more”. Line 169 should put a comma after “men do, indeed, show lower sympathy than women across”.  Line 171 should say “This finding is in part”.  Line 173 should say “The explanations underlying much of these differences are unknown”.  Line 175 should put a period after reference 62 and start the next sentence “For example,”.  Lines 178-179 should say “Implicit theories of emotion implicates an understanding of one’s emotions and the ability to control those emotions.”  Line 180 should say “and, therefore, naturally encourage”.  Line 181 should say “norms): Western cultures emphasize independences, and they encourage self”.  Line 183 should put commas around “therefore”.  Line 184 should say “show high levels of empathy…theories of emotion compared”. 

I provide the aforementioned English corrections to help the authors improve that prose.  More importantly, I hope the authors apply the clarity gained from those corrections to the remainder of the paper. 

Author Response

  • Point 1 : I am reviewing “Cross-cultural Comparison of Relationships between Empathy and Implicit Theories (in Chinese and Russians)” for Behavioral Sciences.  I believe that the authors are trying to examine the factors that predict empathy across three samples: Russians, Chinese, and Chinese living in Russia.  However, the reliabilities for the ITE are very low, the samples are different sizes and ages, and the patterns that the authors say are similar in the figures do not look similar at all. 

Response 1: Thank you very much! We have addressed the reviewers’c concerns throughout the text, and revised the manuscript extensively. With respect to the ITE reliability, we respectfully disagree – although the reviewer is not citing specific guidelines or cut-offs for ‘very low’ reliability (one can assume the classic guidelines of Nunnaly, 1967), reliabilities in the >.65-70 range are not unusual for research instruments and in particular when new concepts or multidimensional concepts are being studied and measured. We acknowledge that the goal of the study was not to use that for individual diagnostic purposes (see Streiner D. L. Starting at the beginning: an introduction to coefficient alpha and internal consistency // Journal of Personality Assessment. 2003. V. 80. № 1. P. 99–103)

  • Point 2: The authors should also evaluate and display the many dependent variables across gender and cultural group in a table or figure. 

Response 2: Thank you for this suggestion. We have included a description of study variables in the Methods section and expanded their contextualization and definitions in the Introduction. We now provide all descriptives regarding gender and age in the manuscript (Table 1).

  • Point 3: I will make suggestions that may help the authors simplify their analyses to evaluate whether the different samples show similar relations between predictors of empathy and empathy.  The prose of the paper suffers from the lack of clarity in many places, and I will provide several examples for the authors that they can use to fix the English throughout the paper.
  • The ITE reliabilities are woefully low.  The authors could re-run the analyses for the entire sample, created from combining all three samples, and remove items that lower the alpha levels below .7 for the ITE and any other measures.  The reliabilities should be provided for every measure in the study.  The authors should evaluate measures with reliabilities at or above .7 across gender and sample type (3) and they should statistically control for age.  The authors should also statistically control for the sample size if possible.  The authors should examine the measures across gender and sample type controlling for age and put that information in tables or figures.

Response 3: Please see our response above. The ‘guidelines’ for interpretation of coefficient alpha are just that – guidelines, are the ones most frequently cited in the field, unfortunately, are remarkably old and arbitrary and neglect the complexity of some of the traits where adequate domain representation necessarily interferes with shared variance. The original questionnaire is 4 items long, and keep that in mind is pivotal for interpretation of alpha, as it is a function of  both the magnitude of inter-item correlations and questionnaire/assessment length.  It is unclear whether the reviewer is suggesting we should abandon the data – we note that we exert systematic effort in adjusting for age and gender by using propensity score matching. We also note that age does not moderate the effects observed in the study. We provide details on all of these in the Methods. Acknowledging (small but statistically significant) differences in gender ratios and age distributions between the recruited samples, we have attempted to at least partially mitigate that by using propensity score matching and selecting samples that were closest by gender and age distribution to the group of Chinese living in China (e.g., the Russian sample was reduced to N=400 to enable age and gender distribution comparability). We now also provide information on how many participants filled out the assessments in a paper-and-pencil format, and how many – via online forms. n Russia participants were recruited via Moscow State University or via the Internet flyer advertising the study. We did not compensate the Russian participants. The Chinese participants consisted of those tested in person and online through an in-vitation of their fellow students in the Chinese community. After completing the questionnaire, each Chinese participant received 5-10 yuan to compensate for the lost time.

We are open to other suggestions, if the above does not address the reviewer’s concerns sufficiently.

  • Point 4: Rather than conducting several path model analyses across the three samples, the authors should consider conducting regression analyses across the three samples with empathy as the criterion and the same variables as predictors. 

Response 4: We think there might be a misunderstanding regarding the mixture modeling/latent class analysis model formulation in the study. There was no goal to predict empathy in this study – although implicitly establishing multiple homogenous subgroups accounts for variation in indicator variables. We present detailed modeling procedures description in the text and the accompanying references.

  • Point 5: The same predictors should be placed in hierarchical regression equations in the same order based on the literature.  In fact, the authors should rework the introduction to set up the order of those variables regardless of the analysis type.  The authors should statistically control for age (put first in the regression analysis) and they should control for sample size if possible.  The current modeling approach is fine if the authors can control for sample age and, hopefully, sample size.  I think that sample size is difficult to control statistically, but the results may be different or similar simply because the samples differ so dramatically in size.  The authors could randomly pull participants from the larger samples to make sure that all three samples are similar in size; I do not like this approach if sample size can be controlled statistically.    

Response 5: Thank you for this suggestion. We respectfully disagree that hierarchical regression modeling here is appropriate given the aims of the study, which guided the selection of the specific analytical procedures. We have broadened the sample via additional recruitment, and utilized it to perform matching and arrive at more comparable (age and gender distribution-wise) samples. We acknowledge this as a limitation. We note that matching procedures evened out the resulting sample Ns, and that the results are nonetheless fairly robust to sample (and subsample) selection at this N.

  • Point 6: The current analyses with median splits and models with multiple factors that change across the three samples is super confusing.  In addition, the figures that the authors display and refer to as similar do not look similar at all.  Instead, the authors can present the Beta weights for the factors in the regression equation and a comparison of those Beta weights can be made and conclusions can be drawn.

Response 6: Please see our response above. We present detailed data regarding the predicted probabilities of endorsement of categories of personality variables in two classes, and happy to provide additional mixture modeling outputs, but we invite the reviewer to explicate the analysis plan as it maps onto the study goals to guide us in a further revision, if necessary. 

  • Point 7: The English is very confusing in several spots in the paper.  I will start with the abstract. Lines 14-17 should say “differences. Chinese living in China showed higher cognitive and affective empathy levels than Chinese living in Russia, and affective empathy was higher overall in women than in men. Finally, women in the combined Chinese sample showed higher affective and cognitive empathy levels as well as incremental implicit theories than men in the combined Chinese sample.”  Was this last comparison actually made?

Response 7: We have adjusted this and the below commentaries. We thank the reviewer for the detailed and structured feedback on the manuscript, and hope that extensive revisions and recruitment of additional external expertise (i.e., language editing) have addressed the reviewer’s concerns. We are happy to follow-up on the unresolved issues, if there are any.

  • Point 8: Line 25 should put commas around “therefore” and line 27 should put a period after “collective” and start the new sentence with “Collectivistic cultures”.  Line 30 should start “Importantly, cultural values are susceptible”.  Line 35 should put a period after “self-construal” and capitalize “Interdependence” for the start of the next sentence.  Line 37 should say “social relationships, with a realization”.  Lines 39-43 should say “Wang and Conway [35] asked participants to recall 20 events and they found the American participants provided more….than the Chinese participants, whereas the Chinese participants were likely to recall…and they detailed social interactions.”

Lines 44-45 should say “for people sharing occupied territory and language as well as a rich set of…”  The next sentence should start “For example, Western and Eastern”.  Line 49 should say “differ in the way individuals”.  Line 59 should say “and it is an important….determining social distancing…”.  Line 63 should say “During a previous epidemic”.  Line 64 should say “but they also reported more”.  I have no idea what lines 65 and 66 are saying to the point that I cannot recommend help.  Lines 67-68 should say “behavior is influenced substantially by culture, which particularly includes empathy and implicit theories…”.  Line 71 should say “multifaceted state and trait”.  Line 72 should delete the text after reference 17 in that sentence.  Line 74 should say “whereas other individuals”.  Line 75 should say “shapes, ranging from emotional”.  Line 78 should put the e.g., information in parentheses.  I have no idea what is being said n lines 80-81.  Line 85 should say “those individuals who score high”.  I have no idea what is being said in lines 84-87. 

I believe the “&” on line 93 should be “and”.  Line 94 should say “studied coping strategies for women experiencing normal pregnancies in Japan and America.”  Line 96 should say “predictor for decisions to social…Russian participants but not in Chinese participants”.  Line 98 should say “students who either”.  Line 99 should say “China. By measuring a wide”.  I have no  idea what is being said in lines 99-100 and in line 105.  I have no idea what is being said in lines 109-113.  Line 114 is confusing.  Line 115 should say “with the following Big Five traits: Conscientiousness, Agreeableness, and Openness”.  Line 119 should say “understand other individuals to explain”.

Line 137 should say “ Implicit…are a domain of individual differences….that is studied less frequently than the domain of empathy or…”  Line 143 should put parentheses around the e.g., information.  Line 151 cannot start with an acronym.  Line 151 says “those of ability” and this phrasing is very confusing. Line 153 should say “whereas constant theorists assume”.  Line 156 should say “but they are not related”. Lines 158 and 159 should put a period after the 55 reference and capitalize “Yet” to start a new sentence.  Lines 160-162 should say “knowledge, no research study has directly examined the pivotal….regulation, and its relationship to empathy.”

Line 163 should say “domain than …is the domain of gender differences”.  Line 163-164 should say “work, men’s thinking was characterized”.  Line 165 should say “outside world, whereas women’s thinking, in contrast, was more”. Line 169 should put a comma after “men do, indeed, show lower sympathy than women across”.  Line 171 should say “This finding is in part”.  Line 173 should say “The explanations underlying much of these differences are unknown”.  Line 175 should put a period after reference 62 and start the next sentence “For example,”.  Lines 178-179 should say “Implicit theories of emotion implicates an understanding of one’s emotions and the ability to control those emotions.”  Line 180 should say “and, therefore, naturally encourage”.  Line 181 should say “norms): Western cultures emphasize independences, and they encourage self”.  Line 183 should put commas around “therefore”.  Line 184 should say “show high levels of empathy…theories of emotion compared”. 

I provide the aforementioned English corrections to help the authors improve that prose.  More importantly, I hope the authors apply the clarity gained from those corrections to the remainder of the paper. 

Response 8: Thank you very much!

Round 2

Reviewer 1 Report

This version of the presented manuscript explains better the study and the results than the previous one.

Author Response

  • The Introduction is still relatively long, especially compared to the Discussion. I strongly suggest to get rid of various superfluous details here and make sure to succinctly summarise the basics that the reader needs to know to understand the background, the motivation, and the hypotheses of the study.
  • Thank you for this suggestion. We have revised the Introduction, shortened it, and instead expanded the Discussion.

The Discussion, on the other hand, could be further expanded and clarified to explain to the reader not only what was found, but also what the findings mean more broadly and what assumptions they are based on. Also consider specific limitations of your study (not just relatively general ones like this already given). For example, right now, no mention is made of the fact that the group of Chinese living in Russia consists almost exclusively of students. These are people willing to move to a foreign country for the sake of academic education which means they are likely not representative of the general population. Therefore any differences observed between them and the other groups could also be driven by this factor. Would the results still present when only students were included in the analysis? 

  • We thank the reviewer for this suggestion. Following this recommendation, we added to the Results an entire subsection (3.6) dedicated to the analyses within student samples. We report that the results are robust to this sample restriction and identify similar high- and low-empathy classes. However, we do not report on the effects specific to students, particularly as related to the association between implicit theories of emotion and cognitive empathy. These results are not discussed within the proper context, as recommended by the reviewer.
  •  

Reviewer 2 Report

I thank the authors for their responses. It is clear that they have made an effort to improve the critical points identified in my earlier assessment and the manuscript has benefitted from this.

The following aspects, in my opinion, should still be addressed:

  • The Introduction is still relatively long, especially compared to the Discussion. I strongly suggest to get rid of various superfluous details here and make sure to succinctly summarise the basics that the reader needs to know to understand the background, the motivation, and the hypotheses of the study.
  • The Discussion, on the other hand, could be further expanded and clarified to explain to the reader not only what was found, but also what the findings mean more broadly and what assumptions they are based on. Also consider specific limitations of your study (not just relatively general ones like this already given). For example, right now, no mention is made of the fact that the group of Chinese living in Russia consists almost exclusively of students. These are people willing to move to a foreign country for the sake of academic education which means they are likely not representative of the general population. Therefore any differences observed between them and the other groups could also be driven by this factor. Would the results still present when only students were included in the analysis? 

Author Response

The following aspects, in my opinion, should still be addressed:

  • The Introduction is still relatively long, especially compared to the Discussion. I strongly suggest to get rid of various superfluous details here and make sure to succinctly summarise the basics that the reader needs to know to understand the background, the motivation, and the hypotheses of the study.
  • Thank you for this suggestion. We have revised the Introduction, shortened it, and instead expanded the Discussion.
  • The Discussion, on the other hand, could be further expanded and clarified to explain to the reader not only what was found, but also what the findings mean more broadly and what assumptions they are based on. Also consider specific limitations of your study (not just relatively general ones like this already given). For example, right now, no mention is made of the fact that the group of Chinese living in Russia consists almost exclusively of students. These are people willing to move to a foreign country for the sake of academic education which means they are likely not representative of the general population. Therefore any differences observed between them and the other groups could also be driven by this factor. Would the results still present when only students were included in the analysis? 
  • We thank the reviewer for this suggestion. Following this recommendation, we added to the Results an entire subsection (3.6) dedicated to the analyses within student samples. We report that the results are robust to this sample restriction and identify similar high- and low-empathy classes. However, we do not report on the effects specific to students, particularly as related to the association between implicit theories of emotion and cognitive empathy. These results are not discussed within the proper context, as recommended by the reviewer.
  •  

Reviewer 4 Report

I am reviewing for a second time “Cross-cultural Comparison of Relationships between Empathy and Implicit Theories (in Chinese and Russians)” for Behavioral Sciences.  I like the fact that the authors made the samples equivalent based on the proportion of demographic variables, which allows the samples to be compared even though they are not equivalent in terms of sample size.  I will describe several problems with the way the statistics are presented.  In addition, the conclusions should not precede the limitations and the English of the paper needs to be improved in many places.

The authors should not code the groups with numbers (Groups 1, 2, and 3).  This method of presentation only confuses the reader and the reader has do mental gymnastics to figure it out each time and the authors often present the labels.  The solution is simple; simply present the labels with Russians, Chinese in Russia, and Chinese in China.   

The authors incorrectly evaluate the different groups for men and women and put that information in a section labeled gender differences.  Gender differences occur when a difference is found across gender or an interaction occurs involving gender.  If the authors found the latter, they can present that statistic and then break down the groups by gender, but they should not make this broken-down comparison without a reason.  From a visual inspection, I do not think the authors show an interaction, so they should simply compare the 3 groups or the 2 groups when they decide to combine 2 groups.  In reference to this move, the authors need to justify the reason they are combining 2 groups.  Typically, the three groups are compared for differences and then follow up comparisons can be made between individual groups or any single group compared to combinations of groups, but a reason needs to be provided for that combination.  Again, the authors can compare gender and they can compare the 3 groups, but they should not compare breakdowns of those groups unless they find a significant Gender by Group interaction.

I will make grammar suggestions for clarity later in the Correlational analysis section.  The authors say they “preferred” or “chose” many times in reference to the best models.  It does not matter what the authors prefer or would like to choose.  Instead, the data reveal that one model is superior to another model.  The authors should phrase it that way.  I will provide one example the first time it appears in the paper when I provide grammar suggestions.  The authors primarily say model class (x) but they also say model class x.  Parentheses hide information from the regular text, so the authors should say model class x.   In the Class-specific correlational analyses, the authors provide findings that are nearly the same as the results in the Correlational analysis section.  The authors need to make this point.  The authors also need to make the point that they do not find a positive relation between ITE and perspective taking in the Class-specific correlational analyses. 

As I stated previously, conclusions mean an end to the Discussion and the paper, so limitations cannot follow.  The authors need to describe the process that will provide the methods to capture the distinction between trait and state empathy.

As for the English, the authors need to work hard to improve the clarity of their presentation.  Line 11 should say “would differ” as in hypothetical, not future tense.  The same is true on line 13.  Line 16 should say “compare 2 groups of…those individuals living in China and those individuals living in”.  Line 18 should replace the semicolon with a comma, and the second semicolon with “, and”.  Lines 20-21 should say “reduced to a final sample of Russians (N = 400), a sample of Chinese living… and a sample of Chinese living”.  Lines 24-26 should say “Russian women reported…than Russian men, whereas Chinese women…theories of emotion than Chinese men.”

Line 39 should say “recently been noted”.  Line 43 should delete “furthering”.  Lines 46-50 should say “showed that American participants asked to recall 20 events frequently provided responses detailing…and events, whereas Chinese participants were likely to recall…social importance, as well as detailed”.  The authors should avoid excessively long sentences like this one. 

Line 52 should say “30 years…Soviet Union, previously”.  Lines 54-55 should say “relationships is shifting in that they are somewhat reduced, but they are becoming”.  Line 57 should say “assume a great degree of”.  Line 59 should say “China and Russia were sown to score 20 and 39 out of 100 points on individualism, respectively”. 

Line 61 should say “living for people who are related only via”.  Line 62 should say “language, as well as a wide”.  Line 63 should say “phenotypes.  For example, Western and Eastern”.  Line 67 should say “two sides agreed that”.  Line 68 should say “by a great degree of”.  When the authors provide a comparative (e.g., more, greater, faster), they have to provide a comparison to make sense unless only two groups exist for comparison like women and men.

Line 74 saying “when faced with positive emotions” should be moved to the end of the sentence for best flow.  Line 81 should say “and the way people”.  Line 83 should say “is influenced substantially by cultural factors”.  Line 86 should say “implicates understanding of one’s emotions, and the perceived”.  Line 89 should say “norms). The Western”.  Line 91 should remove the “therefore” from the sentence.  Line 92 should say “were likely to show higher levels of”.  Line 93 should say “the same premises is true”.  Line 97 should say “typologies, using models with relatively”. 

Lines 101-102 should replace the semicolons with commas.  Lines 113 and 114 are so confusing that I cannot help the authors.  Lines 117 and 118 should say “those individuals”.  Line 118 should clarify “This…” something.  What is consistent with the other something?  …?”  Line 131 should use an “and”, not an “&”.  Line 135 should delete the “For example”. 

Line 146 should say “in the culturally-sensitive”.  Line 149 should say “and it was used in the current study.”  Line 162 should not begin a sentence with an acronym.  Commas should be placed around “therefore”.  Line 162 should say “analogously to theories of ability”.  Line 164 should delete “the” before “constant theorists”.  Line 169 should put a period before “Interestingly” for a new sentence.  Line 170 should say “but they are not related”.  Line 172 should say “high effort” and lines 173-174 should say “low levels of…experiencing few…and additional positive emotions”.  Line 178 starts “This” but this what?  Line 182 should say “We anticipated that…would differ”.  Line 185 should put commas around “therefore”.

Line 189 should say “These findings”.  Line 193 should start the sentence after the reference and say “For example, during”.  Line 212 should say “would”.  Line 215 should say “Russia would show”.  Line 216 should say “between Russian participants and Chinese participants”.  Line 219 should say “empathy would”.  Line 221 should delete “a” between “deserves” and “special”.  Line 230 could benefit from “because” instead of “since”.  Line 232 should say “the number of clusters”. 

Line 234 should say “In the current study, we utilized”.  Line 235 should say “who provided similar”.  Line 239 should say “the sample would contain”.  Line 242 should say “would manifest”.  The colon or semicolon on 242 (cannot see it now due to my marks) should be replaced with a comma.  Line 248 should be connected with Line 247 to avoid a one-line paragraph.  Line 248 should say “400 of 523 participants who took part”.  The following lines should remove the coding with Groups 1, 2, and 3, which is very confusing for the reader and puts on them an additional and unnecessary cognitive load. 

Line 266 should say “those individuals”.  Line 269 should say “Ethics Committee of Psychology Faculty from Moscow State University”.  Line 274 should say “participants were”.  Line 276 should say “assessment was based”.  The reliability of the ITE was terrible for Chinese living in China.  The authors should tackle the reason for this poor result in the Discussion or right here.  Line 282 should say “The Questionnaire of….[18] was used for the Chinese”. 

Lines 292 and 293 should list the two types of scales after the colon and the use 2 new sentences to describe those scales.  Line 301 should say “analyses…comparisons”.  Line 307 should say “the p-value”.  Line 308 should replace the comma with “was”.  Line 311 should put “would” before “corresponded”. 

Lines 357 should say “found positive relationships between ITE and both perspective -taking…), and online simulation”.  I understand that the authors want to use preferred, and they can say that one model is preferred over another model, but they cannot say they preferred it.  The authors also say better and parsimonious, but I think they mean “as a more parsimonious”.  The authors should say that they data or results showed a preference for one model over another model. 

Lines 427-428 should say “that empathy was notably higher in a more collectivistic China in comparison with a more individualistic Russia.”  Line 428 should put a comma after “ITE”.  Lines 429-431 are very confusing.  What point is trying to be made?  Line 433 should say “appear to affect levels of empathy”.  Lines 435-436 make no sense.  The authors are comparing Chinese to high affective empathy.  Line 448 starts “It concerns not only”, but what concerns?  Lines 456-460 provides a very long and complicated sentence that should be described in at least 2 simple sentences.  Lines 461-464 should say “third hypothesis because the analyses of the latent classes provide ne and incremental…analysis. In particular, the content analyses demonstrate a higher….domains by the Chinese in comparison”.         

Lines 466-470 should say “Chinese samples. These results may be attributable to…(ISE), which was developed…medical students.  Therefore, the scale may have biased the domain…occupations, with high overall empathy”.  Line 473 should say “levels of affective…empathy, as well as the adoption”.  Line 479 should put commas around “therefore” and line 482 should be much more specific than saying “thing” twice.  Lines 495-496 should say “These findings suggest…in those individuals who live….as negative relationships”.  Lines 502 should say “in the Russian sample in comparison to the Chinese sample, which can be…more aligned with the individualistic West, and China as more collective.  Line 516 should say “found to report higher”. 

Author Response

The authors should not code the groups with numbers (Groups 1, 2, and 3).  This method of presentation only confuses the reader and the reader has do mental gymnastics to figure it out each time and the authors often present the labels.  The solution is simple; simply present the labels with Russians, Chinese in Russia, and Chinese in China.   

We thank the reviewer for his thorough reading of our manuscript and the many suggestions we have incorporated. For example, this particular recommendation has been implemented throughout.

The authors incorrectly evaluate the different groups for men and women and put that information in a section labeled gender differences.  Gender differences occur when a difference is found across gender or an interaction occurs involving gender. 

We have proceeded following the reviewer’s recommendations. Thus, we now report on the results of a MANOVA that established a statistically significant interaction with sex. This fact, as well as growing literature documenting similar effects, justifies the inclusion of this section, in our mind.

If the authors found the latter, they can present that statistic and then break down the groups by gender, but they should not make this broken-down comparison without a reason.  From a visual inspection, I do not think the authors show an interaction, so they should simply compare the 3 groups or the 2 groups when they decide to combine 2 groups. In reference to this move, the authors need to justify the reason they are combining 2 groups.  Typically, the three groups are compared for differences and then follow up comparisons can be made between individual groups or any single group compared to combinations of groups, but a reason needs to be provided for that combination.  Again, the authors can compare gender and they can compare the 3 groups, but they should not compare breakdowns of those groups unless they find a significant Gender by Group interaction.

We have specifically compared three groups in full accordance with the study goals and hypotheses. That is the reason behind the specific analysis plan implemented for the study. We thank the reviewer for the suggestion, and believe all the comparisons are directly justified in the manuscript.

I will make grammar suggestions for clarity later in the Correlational analysis section.  The authors say they “preferred” or “chose” many times in reference to the best models.  It does not matter what the authors prefer or would like to choose.  Instead, the data reveal that one model is superior to another model.  The authors should phrase it that way.  I will provide one example the first time it appears in the paper when I provide grammar suggestions.  The authors primarily say model class (x) but they also say model class x.  Parentheses hide information from the regular text, so the authors should say model class x.   In the Class-specific correlational analyses, the authors provide findings that are nearly the same as the results in the Correlational analysis section.  The authors need to make this point.  The authors also need to make the point that they do not find a positive relation between ITE and perspective taking in the Class-specific correlational analyses. 

Thank you, this has been added to the text. We also added section 3.6. that details findings in a sample restricuted to students.

As I stated previously, conclusions mean an end to the Discussion and the paper, so limitations cannot follow.  The authors need to describe the process that will provide the methods to capture the distinction between trait and state empathy.

We have modified the manuscript to include a discussion of the differences between empathy as trait and state. Specifically, in expanding the Discussion, we now state that “..the emergence in student subsamples of students of the relationship between incremental ITE and cognitive empathy is specific to the high-empathy class, .. which supports the framework of viewing it as a developing process rooted in cognition. .. We view the relationship between ITE and affective empathy in the low-empathy class similarly. High levels of empathy are achieved via cognitive rather than affective mediation, consistent with the idea of “empathy modeling as a higher psychological function” (Vygotsky).

As for the English, the authors need to work hard to improve the clarity of their presentation.  Line 11 should say “would differ” as in hypothetical, not future tense.  The same is true on line 13.  Line 16 should say “compare 2 groups of…those individuals living in China and those individuals living in”.  Line 18 should replace the semicolon with a comma, and the second semicolon with “, and”.  Lines 20-21 should say “reduced to a final sample of Russians (N = 400), a sample of Chinese living… and a sample of Chinese living”.  Lines 24-26 should say “Russian women reported…than Russian men, whereas Chinese women…theories of emotion than Chinese men.”

Line 39 should say “recently been noted”.  Line 43 should delete “furthering”.  Lines 46-50 should say “showed that American participants asked to recall 20 events frequently provided responses detailing…and events, whereas Chinese participants were likely to recall…social importance, as well as detailed”.  The authors should avoid excessively long sentences like this one. 

Line 52 should say “30 years…Soviet Union, previously”.  Lines 54-55 should say “relationships is shifting in that they are somewhat reduced, but they are becoming”.  Line 57 should say “assume a great degree of”.  Line 59 should say “China and Russia were sown to score 20 and 39 out of 100 points on individualism, respectively”. 

Line 61 should say “living for people who are related only via”.  Line 62 should say “language, as well as a wide”.  Line 63 should say “phenotypes.  For example, Western and Eastern”.  Line 67 should say “two sides agreed that”.  Line 68 should say “by a great degree of”.  When the authors provide a comparative (e.g., more, greater, faster), they have to provide a comparison to make sense unless only two groups exist for comparison like women and men.

Line 74 saying “when faced with positive emotions” should be moved to the end of the sentence for best flow.  Line 81 should say “and the way people”.  Line 83 should say “is influenced substantially by cultural factors”.  Line 86 should say “implicates understanding of one’s emotions, and the perceived”.  Line 89 should say “norms). The Western”.  Line 91 should remove the “therefore” from the sentence.  Line 92 should say “were likely to show higher levels of”.  Line 93 should say “the same premises is true”.  Line 97 should say “typologies, using models with relatively”. 

Lines 101-102 should replace the semicolons with commas.  Lines 113 and 114 are so confusing that I cannot help the authors.  Lines 117 and 118 should say “those individuals”.  Line 118 should clarify “This…” something.  What is consistent with the other something?  …?”  Line 131 should use an “and”, not an “&”.  Line 135 should delete the “For example”. 

Line 146 should say “in the culturally-sensitive”.  Line 149 should say “and it was used in the current study.”  Line 162 should not begin a sentence with an acronym.  Commas should be placed around “therefore”.  Line 162 should say “analogously to theories of ability”.  Line 164 should delete “the” before “constant theorists”.  Line 169 should put a period before “Interestingly” for a new sentence.  Line 170 should say “but they are not related”.  Line 172 should say “high effort” and lines 173-174 should say “low levels of…experiencing few…and additional positive emotions”.  Line 178 starts “This” but this what?  Line 182 should say “We anticipated that…would differ”.  Line 185 should put commas around “therefore”.

Line 189 should say “These findings”.  Line 193 should start the sentence after the reference and say “For example, during”.  Line 212 should say “would”.  Line 215 should say “Russia would show”.  Line 216 should say “between Russian participants and Chinese participants”.  Line 219 should say “empathy would”.  Line 221 should delete “a” between “deserves” and “special”.  Line 230 could benefit from “because” instead of “since”.  Line 232 should say “the number of clusters”. 

Line 234 should say “In the current study, we utilized”.  Line 235 should say “who provided similar”.  Line 239 should say “the sample would contain”.  Line 242 should say “would manifest”.  The colon or semicolon on 242 (cannot see it now due to my marks) should be replaced with a comma.  Line 248 should be connected with Line 247 to avoid a one-line paragraph.  Line 248 should say “400 of 523 participants who took part”.  The following lines should remove the coding with Groups 1, 2, and 3, which is very confusing for the reader and puts on them an additional and unnecessary cognitive load. 

Line 266 should say “those individuals”.  Line 269 should say “Ethics Committee of Psychology Faculty from Moscow State University”.  Line 274 should say “participants were”.  Line 276 should say “assessment was based”.  The reliability of the ITE was terrible for Chinese living in China.  The authors should tackle the reason for this poor result in the Discussion or right here.  Line 282 should say “The Questionnaire of….[18] was used for the Chinese”. 

Lines 292 and 293 should list the two types of scales after the colon and the use 2 new sentences to describe those scales.  Line 301 should say “analyses…comparisons”.  Line 307 should say “the p-value”.  Line 308 should replace the comma with “was”.  Line 311 should put “would” before “corresponded”. 

Lines 357 should say “found positive relationships between ITE and both perspective -taking…), and online simulation”.  I understand that the authors want to use preferred, and they can say that one model is preferred over another model, but they cannot say they preferred it.  The authors also say better and parsimonious, but I think they mean “as a more parsimonious”.  The authors should say that they data or results showed a preference for one model over another model. 

Lines 427-428 should say “that empathy was notably higher in a more collectivistic China in comparison with a more individualistic Russia.”  Line 428 should put a comma after “ITE”.  Lines 429-431 are very confusing.  What point is trying to be made?  Line 433 should say “appear to affect levels of empathy”.  Lines 435-436 make no sense.  The authors are comparing Chinese to high affective empathy.  Line 448 starts “It concerns not only”, but what concerns?  Lines 456-460 provides a very long and complicated sentence that should be described in at least 2 simple sentences.  Lines 461-464 should say “third hypothesis because the analyses of the latent classes provide ne and incremental…analysis. In particular, the content analyses demonstrate a higher….domains by the Chinese in comparison”.         

Lines 466-470 should say “Chinese samples. These results may be attributable to…(ISE), which was developed…medical students.  Therefore, the scale may have biased the domain…occupations, with high overall empathy”.  Line 473 should say “levels of affective…empathy, as well as the adoption”.  Line 479 should put commas around “therefore” and line 482 should be much more specific than saying “thing” twice.  Lines 495-496 should say “These findings suggest…in those individuals who live….as negative relationships”.  Lines 502 should say “in the Russian sample in comparison to the Chinese sample, which can be…more aligned with the individualistic West, and China as more collective.  Line 516 should say “found to report higher”. 

We have incorporated all of the reviewer’s suggestions. We thank the reviewer for his attention to detail and highlight the fact that the manuscript has been reviewed and edited by a scientist with a native level of proficiency in English. While we assume that certain idiosyncrasies remain (as is always the case when the manuscript is written by a non-English speaker), we stress that we believe they are either resolved or minor.
